# Pretreatment of Ultra-Weak Fiber Bragg Grating Hydrophone Array Based on Cubic Spline Interpolation Using Intensity Compensation

**DOI:** 10.3390/s22186814

**Published:** 2022-09-08

**Authors:** Yandong Pang, Hanjie Liu, Ciming Zhou, Junbin Huang, Hongcan Gu, Zhiqiang Zhang

**Affiliations:** 1Department of Weaponry Engineering, Naval Engineering University, Wuhan 430070, China; 2National Engineering Laboratory for Fiber Optic Sensor Technology, Wuhan University of Technology, Wuhan 430032, China

**Keywords:** fiber-optic sensor, ultra-weak fiber Bragg gratings, cubic spline interpolation, hydrophone

## Abstract

The demodulation algorithm based on 3 × 3 coupler in a fiber-optic hydrophone array has gained extensive attention in recent years. The traditional method uses a circulator to construct the normal path-match interferometry; however, the problem of increasing the asymmetry of the three-way signal to be demodulated is easily overlooked. To provide a solution, we report a pretreatment method for hydrophone array based on 3 × 3 coupler demodulation. We use cubic spline interpolation to perform nonlinear fitting to the reflected pulse train and calculate the peak-to-peak values of the single pulse to determine the light intensity compensation coefficient of the interference signal, so as to demodulate the corrected three-way interference signal. For experimental verification, ultra-weak fiber Bragg gratings (uwFBGs) with reflectivity of −50 dB are applied to construct a hydrophone array with 800 sensors, and a vibratory liquid column method is set up to generate a low-frequency hydroacoustic signal. Compared to the traditional demodulation algorithm based on a 3 × 3 coupler, the pretreatment method can improve the consistency of interference signals. The Lissajous figures show that cubic spline interpolation can improve the accuracy of monopulse peak seeking results by about 1 dB, and intensity compensation can further lead to a much lower noise density level for the interference pulse amplitude—specifically, more than 7 dB at 5~50 Hz—and the signal-to-noise ratio is improved by approximately 10 dB at 10 Hz. The distinct advantages of the proposed pretreatment method make it an excellent candidate for a hydrophone array system based on path-match interferometry.

## 1. Introduction

Fiber-optic sensors have been widely studied in recent years [1] for their advantages of high-sensitivity, anti-electromagnetic interference, and large-scale reuse. Among various types of fiber-optic sensors, FBG-based sensor networks can be configured differently according to individual practical scenarios, and the multiplexing of FBGs has been demonstrated [2], such as vibration sensing based on simultaneous vector demodulation [3], hot spot detection systems [4], Optical Time Domain Reflectometry (OTDR) sensing networks [5], and Distributed Acoustic Sensing (DAS) systems with long sensing distances [6]. Several current applications within aeronautical engineering that use fiber-optic systems include commercial aircraft, unmanned aircraft, international space exploration systems, and so on [7]. As well as within civil engineering, a long-term health monitoring and alarm system of structures based on a number of optical fiber sensing has been established and applied successfully in Wuhu Yangtze River Bridge and the high and steep slope of Shuohuang Railway [8]. For underwater environment applications, large-scale reuse and excellent resolution are the key features of fiber-optic hydrophones, especially in a surface towed array sonar system [9], because increasing the sensor number of arrays can improve the detection range, and good spatial resolution can further enhance the detection accuracy.

In order to achieve the high spatial resolution of quasi-distributed time division multiplexing (TDM) array, researchers generally reduce the grating interval or improve the system sampling rate. In this case, the pulse width should be decreased as much as possible to ensure that the adjacent light pulses do not overlap. Moreover, when ultra-weak fiber Bragg gratings (uwFBGs) are used to form a sensor array, the reflectivity of each uwFBG is between −40 dB and −50 dB [10], so detecting the weak light pulse with high precision and selecting a stable demodulation algorithm of interference signals have an important influence on the system performance. A previous study has proved that algorithms based on a 3 × 3 coupler are able to demodulate uwFBG signals due to the simple architecture and immunity to signal fading [11]. 

However, we use a 3 × 3 coupler to obtain the interference signal. The insertion loss of the circulator will reduce the signal intensity, but on the other hand, the arrival time for the three-way echo pulse to the detector will not be consistent owing to the optical path extension caused by the circulator. Especially when the detection accuracy of fiber-optic delay is inadequate or is difficult to measure, fluctuations in the pulse amplitude will be unpredictable, and the combined effect for the two aspects will further increase the asymmetric results of the three-way signal [3]. Therefore, under the circumstances of optical pulse drift, it is highly likely that the pulse of three-way signals cannot reach the detector at the same time, and the intensity of interference signals will be greatly reduced.

Based on this issue, Ding [12] attempted to demodulate only through Hilbert transform to reduce the influence of the circulator; however, the experimental results show a low signal-to-noise ratio (SNR). In order to eliminate the intensity noise, and subsequently the influence of light intensity disturbance (LID), Zhang [13] reported an improved phase-generated carrier (PGC) demodulation algorithm, and they further proposed an asymmetric division demodulation algorithm based on fundamental frequency mixing [14]. The SNR and distortion ratio of the sensor achieves a gain of 7.77 dB over the traditional algorithm; however, the above methods only apply to PGC and its improved algorithms. For the FBG interference scheme with a 3 × 3 coupler, the non-interferential pulse is used as a reference signal to suppress signal distortion caused by optical intensity [15]. The total harmonic distortion can be decreased by 30 dB, but it cannot suppress the change in interference pulse intensity caused by optical pulse drift. It is worth noting that light intensity based on the 3 × 3 coupler can form an elliptic curve [16]. Inspired by this observation, Qu [17] proposed an ellipse fitting algorithm (EFA) to compensate for the nonlinear distortion. However, when the phase change is less than π due to the mild intrusion, the interference signal cannot form a complete ellipse, which leads to the error of parameter estimation. To solve this problem, Shi [18] proposed a phase-shifted demodulation technique with a 3 × 3 coupler and EFA for the interrogation of interferometric sensors, and their experimental result showed high accuracy and stability for measuring small phase signals. Although this method can perfectly solve many problems resulting from parameter inconsistency, it has high complexity. Focusing on TDM systems, when the sensor spacing magnitude is meter-scale, the sampling rate must reach hundreds of megahertz to achieve more accurate interference signals [19], but for high-speed acquisition systems, the EFA method undoubtedly increases the burden of data processing.

In this paper, we propose an intensity compensation method of path-match interferometer array based on uwFBG reflectors. At first, in order to precisely locate the trajectory for each pulse peak more, cubic spline interpolation is used to resample the distorted echo pulse. Then, based on the peak coordinates of the pulses with no offset, the intensity compensation of the signal offset by the third pulse is made. This scheme can fundamentally solve the amplitude attenuation of interference signals caused by introducing the circulator; therefore, the demodulation stability of interference signals is efficiently improved.

Considering these ideas, in the following sections, firstly, we discuss the source of interference light intensity distortion caused by the circulator and present the mathematical background of the demodulation technique used in our scheme in Section 2. Section 3 is dedicated to the experimental setup to demonstrate the technique. In Section 4, the results of time domain and frequency domain obtained by the proposed method are presented, and the SNR enhancement results at different frequencies are calculated.

## 2. Principle

Figure 1 shows the structure of the path-match interferometric fiber-optic system. The continuous light emitted from a narrow linewidth laser is modulated into pulse light by an acousto-optic modulator (AOM), which is injected into circulator-1 (CIR-1) followed by amplification by an erbium-doped fiber amplifier (EDFA). Then, the pulse enters the uwFBGs array, and the echo pulse train reflected by the uwFBGs is transmitted to CIR-2. The arm length difference of the nonequilibrium interferometer constructed by a Faraday rotating mirror (FRM) is the same with the spacing of uwFBGs; therefore, optical path compensation can be realized in the interferometer by successive reflected pulses, thus completing the pulse interference. Finally, the interference pulse is converted by photoelectric detectors (PDs) into a field programmable gate array (FPGA) module for data acquisition and processing. 

When the 3 × 3 coupler outputs three interference pulses, the three echo pulses will not be transmitted synchronously due to the existence of CIR-1 in channel C-1, as shown in Figure 1. When we select the fixed peak position of echo pulse to obtain interference signals, the intensity of interference signals will be inconsistent for C-1, C-2 and C-3.

In order to satisfy the condition of the 3 × 3 demodulation algorithm, we usually introduce CIR-2 to transmit the optical signal of C-1. As for the specific analysis, the echo pulse train and interference signal are shown in a dotted line diagram. After passing through CIR-2, compared with C-2 and C-3, the pulse of C-1 has delay. In this case, the peak-to-peak amplitude of the three interference signals obtained by fixing the abscissa will be different. In addition, the optical power of the three signals through the coupler is inconsistent. The three-way interference intensity of C-1, C-2, and C-3 can be expressed as
(1)$$\left\{ \begin{array}{l} I_{C-1}=a\left(E_{m}^{2}+E_{k}^{2}\right)+2dE_{m}E_{k}\cos\left(\varphi_{s}\left(t\right)+\varphi_{0}\right)\\ I_{C-2}=b\left(E_{m}^{2}+E_{k}^{2}\right)+2eE_{m}E_{k}\cos\left(\varphi_{s}\left(t\right)+\varphi_{0}+2\pi /3\right)\\ I_{C-3}=c\left(E_{m}^{2}+E_{k}^{2}\right)+2fE_{m}E_{k}\cos\left(\varphi_{s}\left(t\right)+\varphi_{0}+4\pi /3\right) \end{array}\right.,$$
where *E_m_* and *E_k_*, respectively, express the amplitudes of the two fields, and *φ_s_*(*t*) expresses the measured phase change due to the external impact. *φ*_0_ is a constant denoting the initial phases; a represents the DC offset caused by light attenuation of CIR-2; and *a*, *b*, *c*, *d*, *e,* and *f*, respectively, express the variation coefficient due to the pulse delay or inconsistent splitting ratio for C-1, C-2, and C-3. On other hand, the fiber-optic hydrophone array with path-match interferometric belongs to the TDM system; therefore, the time sampling rate has great influence on the discrete pulse signal. When the system sampling rate is low, more intensity noise will be introduced.

In order to solve the above problems, we proposed a pretreatment fiber-optic system with path-match interferometry, as shown in Figure 2. Firstly, cubic spline interpolation was used to make the pulse train smoother, as it can reduce the intensity noise of the interference signal to a certain extent. Secondly, the extreme value of the interference signal was determined by segmented detection, and the period of segmented detection should be obviously longer than that of the signal to be measured; thus, we used C-3 as the reference signal. The attenuation coefficient of C-1 and C-2 could be determined by extreme amplitude comparison, and the intensity of the interference signal of C-1 and C-2 was compensated to complete signal pretreatment. Finally, the modified three-way interference signal was demodulated based on a 3 × 3 coupler.

Different from the FBG reflection spectrum that satisfies Gaussian function distribution [20], there is no relevant literature showing that pulse echo based on path-match interferometry can be described by a fixed mathematical expression. Therefore, in order to reduce the noise of the echo pulse, we applied a cubic spline interpolation algorithm to process the echo pulse. For n + 1 discrete sampling points on the interval [*p*, *q*], the original signal can be interpolated by constructing the cubic polynomial function *s*(*x*) of the sub-interval. Assuming that *s*’’(*x_i_*) = *M_i_*(*i* = 0, 1, …, *n*), *s*’(*x_i_*) = *m_i_*(*i* = 0, 1, …, *n*), we can solve it using the above conditions, and three types of boundary conditions can be expressed, such as
(2)$$\left\{ \begin{array}{l} s\prime \prime \left(x_{0}\right)=f_{0}\prime \prime =M_{0},\;s\prime \prime \left(x_{n}\right)=f_{n}\prime \prime =M_{n}\\ s\prime \left(x_{0}\right)=f_{0}\prime =m_{0},\;s\prime \left(x_{n}\right)=f_{n}\prime \prime =m_{n}\\ s\prime \left(x_{1}+0\right)=s\prime \left(x_{n}+0\right),s\prime \prime \left(x_{1}+0\right)=s\prime \prime \left(x_{n}+0\right) \end{array}\right.,$$
where *s*’(·) and *s*’’(·) represent the first and second derivatives of a polynomial function, respectively. Because the cubic spline interpolation algorithm [19] can be fitted dynamically, it is suitable for processing the echo pulse signal caused by the hydroacoustic wave.

Since each pulse in the echo train represents a sensor, it needs to be segmented according to the sensing distance. For the pulse corresponding to the sensor 1, the interference signal generated by the sensor can be recorded, and the change trajectory of peak value is obtained by using fixed pulse delay. When the phase variation caused by environmental change is greater than π, the extreme value of AC part of the interference signal can be obtained. At the time, when cos(*φ_s_*(*t*) + *φ*_0_) = 1 for C-1, similar to it, we can obtain the extreme values for C-2 and C-3. The maximal detected interference signals for *I*_C-1_, *I*_C-2_, and *I*_C-3_ are expressed as
(3)$$\left\{ \begin{array}{l} {\left(I_{C-1}\right)}_{\max}=a\left(E_{m}^{2}+E_{k}^{2}\right)+2dE_{m}E_{k}\\ {\left(I_{C-2}\right)}_{\max}=b\left(E_{m}^{2}+E_{k}^{2}\right)+2eE_{m}E_{k}\\ {\left(I_{C-3}\right)}_{\max}=c\left(E_{m}^{2}+E_{k}^{2}\right)+2fE_{m}E_{k} \end{array}\right.,$$

When cos(*φ_s_*(*t*) + *φ*_0_) = −1, the minimal detected interference signals for *I*_C-1_, *I*_C-2_, and *I*_C-3_ are
(4)$$\left\{ \begin{array}{l} {\left(I_{C-1}\right)}_{\min}=a\left(E_{m}^{2}+E_{k}^{2}\right)-2dE_{m}E_{k}\\ {\left(I_{C-2}\right)}_{\min}=b\left(E_{m}^{2}+E_{k}^{2}\right)-2eE_{m}E_{k}\\ {\left(I_{C-3}\right)}_{\min}=c\left(E_{m}^{2}+E_{k}^{2}\right)-2fE_{m}E_{k} \end{array}\right.$$

The signals after DC-offset correction can be explained as
(5){(IC-1)′=IC-1−12[(IC-1)max+(IC-1)min](IC-2)′=IC-2−12[(IC-2)max+(IC-2)min](IC-3)′=IC-3−12[(IC-3)max+(IC-3)min]

Owing to the cubic spline interpolation for the echo pulse train in Formula (2), we can obtain a stable peak motion trajectory, and take advantage of it again, to find the extreme value within the segmented interval [*p*, *q*] for *I*’_C-1_, *I*’_C-2_, and *I*’_C-3_. According to Equations (3)–(5), the detected signals *I*’_C-1_ and *I*’_C-2_ for peak-to-peak value (VPP) are given by
(6)$$\left\{ \begin{array}{l} \left(I_{C-1}\right)\prime_{\max}-\left(I_{C-1}\right)\prime_{\min}=4aE_{m}E_{k}\\ \left(I_{C-2}\right)\prime_{\max}-\left(I_{C-2}\right)\prime_{\min}=4bE_{m}E_{k} \end{array}\right.$$

Similarly, the detected VPP signal for *I*’_C-3_ is
(7)$${\left(I_{C-3}\right)}_{\max}-{\left(I_{C-3}\right)}_{\min}=4cE_{m}E_{k}$$

In this way, we can obtain the attenuation coefficient of the interference signal strength due to the pulse offset
(8)$$\left\{ \begin{array}{l} \alpha =\frac{\left(I_{C-1}\right)\prime_{\max}-\left(I_{C-1}\right)\prime_{\min}}{\left(I_{C-3}\right)\prime_{\max}-\left(I_{C-3}\right)\prime_{\min}}=\frac{a}{c}\\ \beta =\frac{\left(I_{C-2}\right)\prime_{\max}-\left(I_{C-2}\right)\prime_{\min}}{\left(I_{C-3}\right)\prime_{\max}-\left(I_{C-3}\right)\prime_{\min}}=\frac{b}{c} \end{array}\right.$$

Then, the three-way updated interference intensity of C-1, C-2, and C-3 can be represented as
(9)$$\left\{ \begin{array}{l} I\prime \prime_{C-1}=\frac{a}{\alpha }\left[2E_{m}E_{k}\cos\left(\varphi_{s}\left(t\right)+\varphi_{0}\right)\right]\\ I\prime \prime_{C-2}=\frac{b}{\beta }\left[2E_{m}E_{k}\cos\left(\varphi_{s}\left(t\right)+\varphi_{0}+2\pi /3\right)\right]\\ I\prime \prime_{C-3}=c\left[2E_{m}E_{k}\cos\left(\varphi_{s}\left(t\right)+\varphi_{0}+4\pi /3\right)\right] \end{array}\right.$$

After compensating for the coefficients, we can use a demodulation algorithm based on a 3 × 3 coupler to obtain the result of the underwater acoustic signal.

## 3. Experimental Setup and Results

The experimental principle of the path-match interferometric fiber-optic hydrophone system is shown in Figure 3. To satisfy the interference conditions in a light source system, the narrow linewidth laser (DFB-M-1550-150-F-10-09MPF-FC/APC) had a bandwidth of 3 kHz and a central wavelength of 1550 nm. The AOM (T-M200-0.1C2J-3-F2S) generated a pulse period of 2 kHz and pulse width of 20 ns. The interference output of the 3 × 3 coupler was composed of three-way parts: C-2 and C-3 routes directly entered the PD, whereas route C-1 reached the PD through port 3 of CIR-2. The detection bandwidth of PDs (KG-200M-APR) is 200 MHz. In order to distinguish each narrow-pulse of sensor, the sampling rate of the data acquisition card (NI PXIe5170R) was set as 250 MSa/s. Finally, cubic spline interpolation and compensation were used to complete the pretreatment in order to achieve stable demodulation.

We used the vibratory fluid column method to simulate the propagation rules of the hydroacoustic wave. The sound pressure field in the liquid column was generated by an excitation table with a linear power amplifier (VT500), and the voltage sensitivity of the accelerometer was 5.76 pC/m·s^−2^. Limited by the output power, the exciter could produce large amplitude only at low frequency; therefore, in order to satisfy the requirement of generating phase variation that is greater than π amplitude, we selected hydroacoustic signals of low-frequency band 5–50 Hz for the experiment. In addition, to obtain the correct signal of standing wave, the inner diameter of the round tube was set as 7.50 cm, and the liquid column height *H* was 18.75 cm. The fiber-optic hydrophone in the experimental system used uwFBGs array with a central wavelength of 1550 nm and reflectivity between –40 and –50 dB [21]. The spacing between adjacent gratings was 5 m. In order to satisfy the matching interference principle, the length difference of the arm of the unbalanced interferometer was also set to 5 m.

Figure 4 shows the three-way pulse signal of uwFBG TDM array with 800 sensors, whose sensing distance is 4 km. By amplifying the partial pulse signal, compared with C-2 and C-3, the C-1 pulse train had an overall offset of *t*_0_, so the amplitude of the C-1 interference signal indexed at the same position was greatly reduced, which verified the theoretical analysis mentioned above.

In our pre-processing algorithm, we first used cubic spline interpolation to fit the echo pulse in order to reduce the intense noise. The original pulse signal and the interpolation result are shown in Figure 5a,b. We can intuitively see that the echo pulse signal is more complete, which proves that the interpolation algorithm is suitable for processing the pulse signal of uwFBGs. For Figure 5a, due to a small number of sampling points, the position corresponding to the pulse peak is prone to deviation, such as the black curve corresponding to channel C-1. However, after interpolation, as shown in Figure 5b, the pulse waveform is significantly improved, and the time position corresponding to the peak is more stable, which is conducive to obtain a more accurate peak change trajectory.

Previous studies have shown that EFA can be used to analyze the nonlinear distortion caused by LID [16]. In the case when the light intensity fluctuates greatly, the edge of the ellipse is more discrete. To verify the effectiveness of cubic spline interpolation, we selected the interference signals of C-2 and C-3 to obtain the Lissajous figure. 

In Figure 6, the blue symbol represents the formation of the original signal, and the red symbol represents the result of cubic spline interpolation. The comparison results show that the cubic spline interpolation lattice is more concentrated than the untreated lattice, hence the intensity noise is lower, indicating that when the interference pulse sampling rate is low, the cubic spline interpolation algorithm can effectively increase the stability of the trajectory of the interference signal. Moreover, it can also explain that when the delay distance caused by the CIR-2 is smaller than the sampling resolution of the acquisition card, we cannot accurately find the position of the peak by estimating the delay, which is an important reason for intensity noise.

We performed an intensity compensation for the three-way signal, and the stable interference signals were obtained through cubic spline interpolation. The original interference signal and the interference signal of compensation are shown in Figure 7a,b. Because of the pulse delay caused by the CIR-2, the peak value of the *I*_C-1_ signal at the same position is greatly attenuated. However, *I*’_C-1_ is improved by linear compensation treatment, and its amplitude is basically consistent with *I*’_C-2_ and *I*’_C-3_. The results prove that linear compensation has a good effect and can effectively reduce the LID generated by pulse delay.

With cubic spline interpolation and linear compensation, the demodulation results are analyzed experimentally. Figure 8a,b shows the comparison of time domain demodulation results for 10 Hz and 30 Hz. Due to the influence of intensity noise, the time domain signals produced obvious distortion, and the demodulation amplitudes were also reduced. The results of algorithm application were effectively improved using the proposed method.

The improvement effect of the proposed method on demodulation signals can be directly explained by time domain signals, but the results cannot be quantified. In order to qualitatively analyze the improvement effect of the proposed algorithm on the integrity of demodulation signals, we compared the power spectral density (PSD) of demodulated signals at 10Hz obtained by three different methods. The PSD obtained by direct demodulation corresponds to Figure 9a, which represents the results obtained without correction. The method proposed in this paper is that the pretreatment of demodulation signals is mainly realized by two steps: First, the original reflected pulse array is processed by cubic spline interpolation (PSD of the demodulation result obtained by this step corresponds to Figure 9b). Figure 9b represents the results obtained by direct compensation without interpolation; Then, the peak trajectory of the smoothed pulse array is obtained, and linear compensation is performed (PSD of the demodulation result obtained in this step corresponds to Figure 9c). Figure 9c represents the results obtained by linear compensation after interpolation. The SNRs of the above three methods were 10.75 dB, 23.21 dB, and 23.31 dB, respectively. The signal without any processing had a low SNR, and the attenuation of the interference signal caused by the pulse delay was the main factor for signal distortion. After linear compensation, the results significantly improved. On the other hand, we analyzed the average values for the PSD of noise when the reference signal was 1 rad/√Hz, which are −36.14 dB, −45.06 dB, and -−46.17 dB, respectively. These could be reduced by 1.11 dB after cubic spline interpolation. Linear compensation can further increase the noise level by about 8.92 dB. The results are also consistent with the results of the Lissajous figure shown in Figure 4, which prove that the proposed algorithm for dealing with the pulse delay problem is highly effective.

Subsequently, demodulation experiments of hydroacoustic signals with different frequencies were conducted for 5 Hz, 10 Hz, 20 Hz, 30 Hz, 40 Hz, and 50 Hz. The qualitative analysis results are shown in Table 1. The average values for the PSD of signals after cubic spline interpolation and linear compensation were greatly improved, and the gain was greater than 7 dB, which verifies the robustness of our proposed algorithm at different frequencies.

## 4. Conclusions

In this paper, to reduce the intensity noise caused by the light delay introduced by the circulator in a 3 × 3 optical path, a linear compensation method based on cubic spline interpolation fitting was proposed. This approach comprises a significant pretreatment method, which can reduce the background noise of low-frequency band by more than 7 dB and improve the SNR by up to 10 dB. The method proposed in this paper only uses extreme values for linear dynamic compensation, without complicated matrix calculation, so that, compared with other compensation algorithms such as EFA, the proposed algorithm is more efficient and has less computation time. With sensor spacing of 5 m with 20 ns echo pulse, it can be immune to the optical path delay within about 2 m caused by the circulator. As a result, the proposed pretreatment method is highly suitable for practical applications in uwFBGs array based on dense TDM systems.

## Figures and Tables

**Figure 1 sensors-22-06814-f001:**
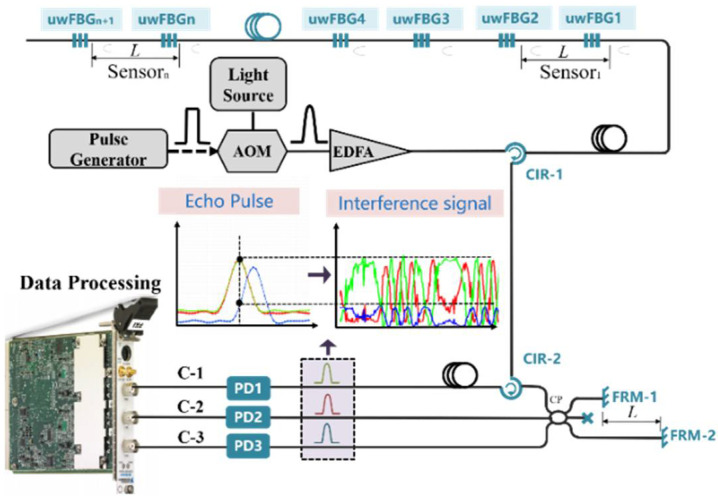
Structure of fiber-optic system with path-match interferometer.

**Figure 2 sensors-22-06814-f002:**
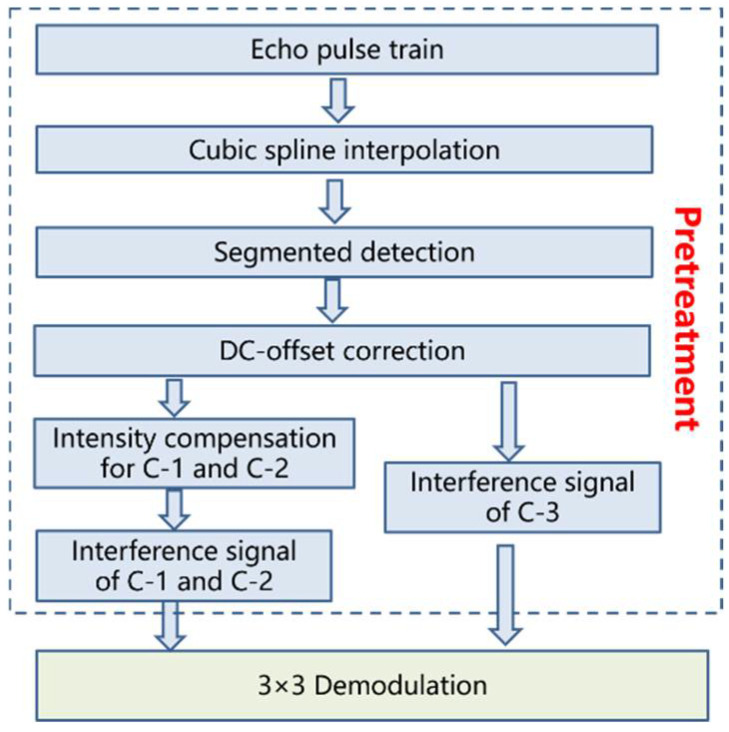
Flowchart of pretreatment method.

**Figure 3 sensors-22-06814-f003:**
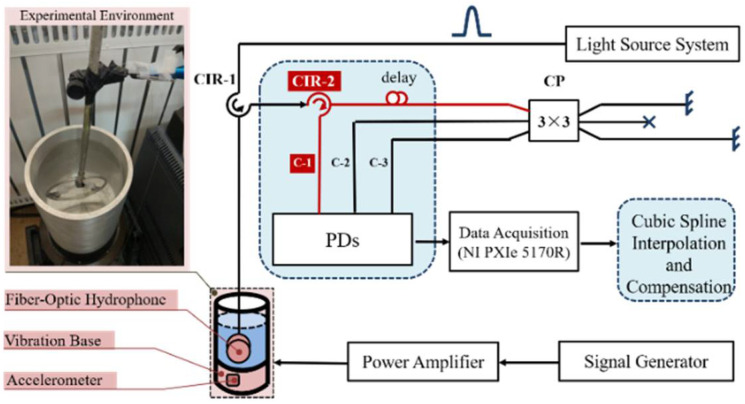
Experimental principle of the path-match interferometric fiber-optic hydrophone system.

**Figure 4 sensors-22-06814-f004:**
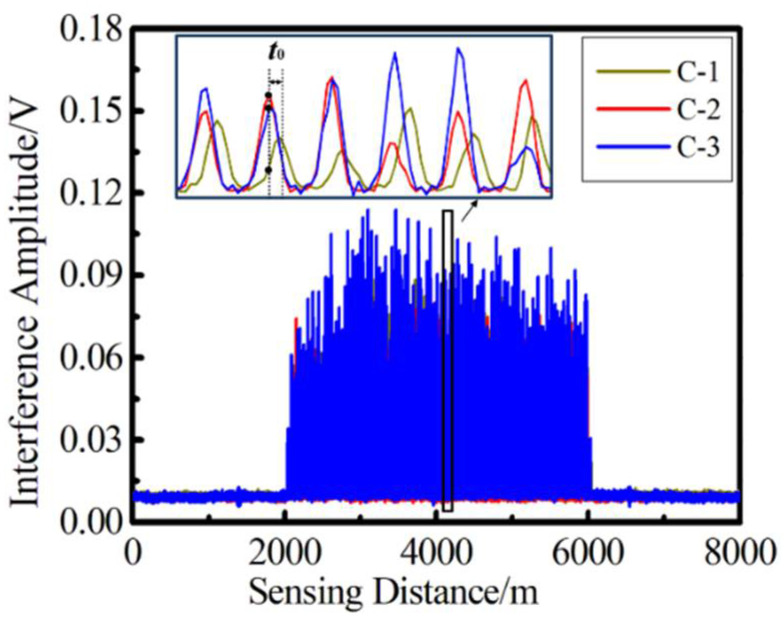
Echo pulse train for three-way signal of uwFBG array with 800 sensors.

**Figure 5 sensors-22-06814-f005:**
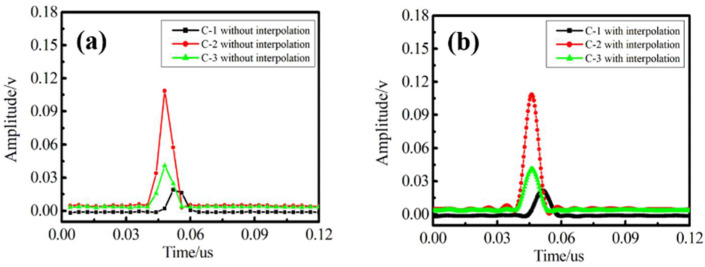
Comparison of echo pulse interpolation results. (**a**), Due to a small number of sampling points, the position corresponding to the pulse peak is prone to deviation, such as the black curve corresponding to channel C-1. (**b**), The pulse waveform is significantly improved, and the time position corresponding to the peak is more stable, which is conducive to obtain a more accurate peak change trajectory.

**Figure 6 sensors-22-06814-f006:**
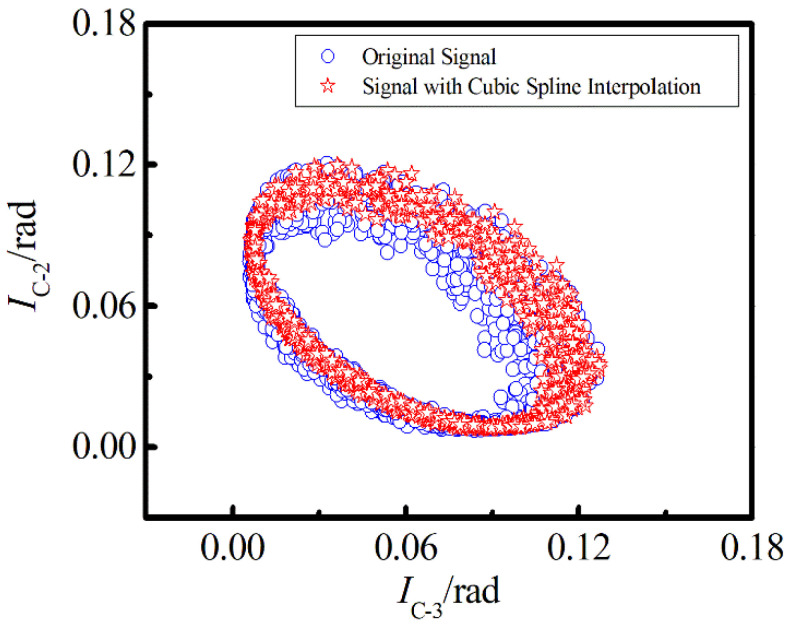
Lissajous figure of interference signals before and after interpolation.

**Figure 7 sensors-22-06814-f007:**
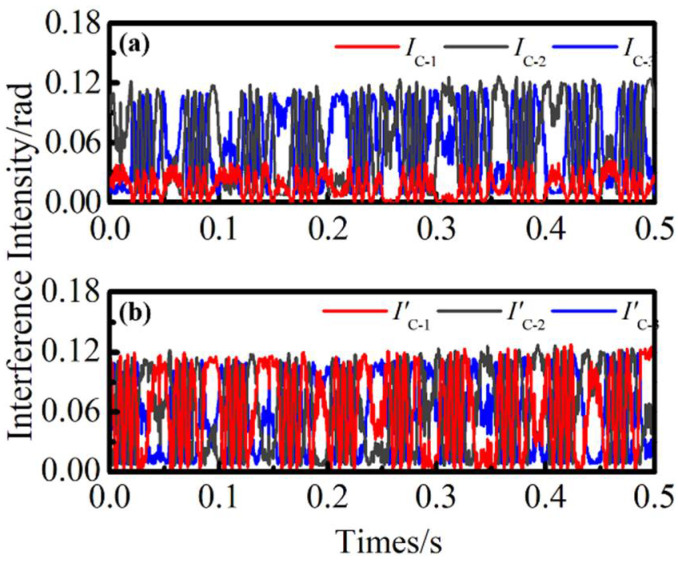
Original interference signal and compensated interference signal.

**Figure 8 sensors-22-06814-f008:**
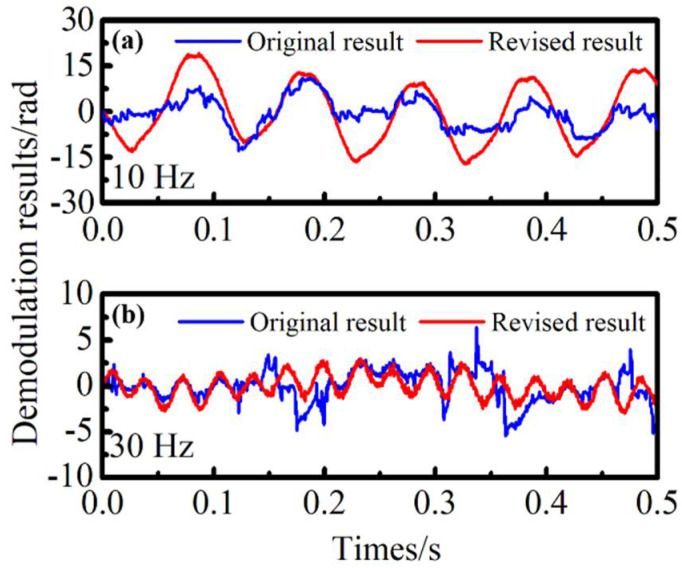
Comparison of time domain demodulation results at 10 Hz and 30 Hz.

**Figure 9 sensors-22-06814-f009:**
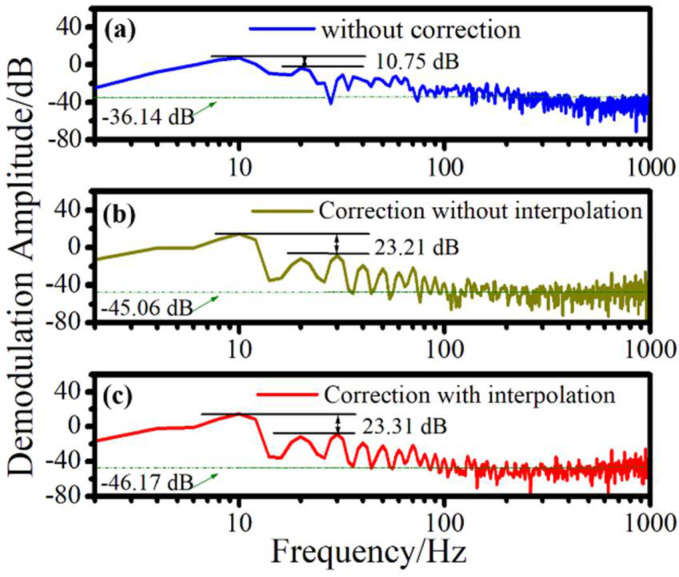
Comparison of PSD with different methods.

**Table 1 sensors-22-06814-t001:** Results of PSD for original and correction at different frequencies.

Frequency/Hz	5	10	20	30	40	50
Original/dB	−38.16	−36.14	−38.64	−37.46	−36.13	−35.50
Correction/dB	−50.35	−46.17	−46.18	−46.73	−47.61	−47.06
Gain/dB	12.19	10.03	7.54	9.27	11.48	11.56

## Data Availability

Not applicable.

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
