# Peer review of "Pretreatment of Ultra-Weak Fiber Bragg Grating Hydrophone Array Based on Cubic Spline Interpolation Using Intensity Compensation"

_sensors, 2022, doi:10.3390/s22186814_

Round 1
Reviewer 1 Report
Comments to the Author
I have reviewed the manuscript “Pretreatment of Ultra-Weak Fiber Bragg Grating Hydrophone Array Based on Cubic Spline Interpolation Using Intensity Compensation” by Yandong Pang, Hanjie Liu, Ciming Zhou, Junbing Huang, Hongcan Gu and Zhiqiang Zhang. This is an interesting and well-organized study and needs some revisions to be published in Sensors. Here are my comments.
1. Please, revise some typos in the text.
2. In the Introduction, please, add some studies dealing with the fiber-optic sensors applications within engineering (e.g., civil engineering).
3. Please, improve the readability of some figures as well as revise some typos (e.g., Figure 7).
Author Response
Comments and Suggestions for Authors
Comments to the Author
I have reviewed the manuscript “Pretreatment of Ultra-Weak Fiber Bragg Grating Hydrophone Array Based on Cubic Spline Interpolation Using Intensity Compensation” by Yandong Pang, Hanjie Liu, Ciming Zhou, Junbing Huang, Hongcan Gu and Zhiqiang Zhang. This is an interesting and well-organized study and needs some revisions to be published in Sensors. Here are my comments.
- Please, revise some typos in the text.
Response: Thank you for the suggestions.
- Some contents of the abstract have been revised, “…and the signal to noise ratio is improved by approximately 10 dB at 10 Hz.”.
- In Page 2, “However, when the phase change is less than π due to the mildly intrusive…”.
- Besides, we revise the full text in detail, and all of revisions to the manuscript have been marked up using the “Track Changes” function.
- In the Introduction, please, add some studies dealing with the fiber-optic sensors applications within engineering (e.g., civil engineering).
Response: We agree with the reviewer’s opinion, and thank you for the suggestions.
(1) We have added the study dealing with the fiber-optic sensors applications within engineering. “Several current applications within aeronautical engineering that use fiber-optic systems include commercial aircraft, unmanned aircraft, international space exploration systems and so on [7]. As well as within civil engineering, a long-term health monitoring and alarm system of structure based on a number of optical fiber sensing have been established and applied successfully in Wuhu Yangtze River Bridge and the high and steep slope of Shuohuang Railway [8].”
- Please, improve the readability of some figures as well as revise some typos (e.g., Figure 7).
Response: Thank you very much for your valuable advices.
- The abscissa of Fig. 7 and Fig. 8 has been modified.
- We marked for different contents in Fig. 7, Fig. 8 and Fig. 9, and added instructions in the caption in the text.
- For improving the readability of figures involving specific quantified results, we explain the content of Fig. 9 in detail. “The improvement effect of the proposed method on demodulation signal can be directly explained by time domain signals, but the results cannot be quantified. In or-der to qualitatively analyze the improvement effect of the proposed algorithm on the integrity of demodulation signal, we compare the power spectral density (PSD) of de-modulated signal at 10Hz obtained by three different methods. The PSD obtained by direct demodulation corresponds to Fig. 9(a), it represents the results obtained without correction. The method proposed in this paper is that the pretreatment of demodulation signal is mainly realized by two steps: first, the original reflected pulse array is processed by cubic spline interpolation (PSD of demodulation result obtained by this step corresponds to Fig. 9(b)), Fig. 9 (b) represents the results obtained by direct compensation without interpolation. Then, the peak trajectory of the smoothed pulse array is obtained and linear compensation is performed (PSD of the demodulation result obtained in this step corresponds to Fig. 9(c)), Fig. 9 (c) represents the results obtained by linear compensation after interpolation. The SNR of above three methods is 10.75 dB, 23.21 dB, 23.31 dB, respectively.”

Reviewer 2 Report
-For the method proposed in this study, is the revised result smoother than the original one? Is smoothness the most essential index for testing and evaluating the methods proposed in this study? For example, as shown in Figure 8?
-In Figure 5, cubic spline interpolation is used to smooth the original data. Why is it cubic spline interpolation? What are the advantages? Can other interpolation algorithms be used? Does the interpolation over fit the original data?
-What does Figure 9 want to illustrate? What is the difference between the three pictures? Please explain in detail so that readers can better understand this study.
-There are too many small and clerical mistakes in the manuscript, such as in abstract "…improved by approximately 10 dB@10 …", in page 2 "…phase change is less then \pi…" and in Figures 7 and 8 "Tims".
Author Response
Comments and Suggestions for Authors
1、For the method proposed in this study, is the revised result smoother than the original one? Is smoothness the most essential index for testing and evaluating the methods proposed in this study? For example, as shown in Figure 8?
Response: Sorry about the confusion.
(1) Smoothness is not the most essential index for testing and evaluating the methods proposed in this study. In our proposed pretreatment method, after cubic spline interpolation, we can obtain more smooth echo pulses, so as to get stable peak trajectory, and the interference of high SNR signal, based on it, we compensate linearly for the interference signal. After completing the above steps to achieve full signal preprocessing, as shown in Fig. 8 (c) is obtained, the result of the compared with the demodulated signal without any processing shown in Fig. 8(a), the signal-to-noise ratio is greatly improved.
2、In Figure 5, cubic spline interpolation is used to smooth the original data. Why is it cubic spline interpolation? What are the advantages? Can other interpolation algorithms be used? Does the interpolation over fit the original data?
Response: Sorry about the confusion and thank you very much for your careful review.
- For FBG reflected pulse, when the pulse width is small enough, the light pulse will have a rising edge and a falling edge due to the limitation of the acouster-optic modulator (AOM), so that the echo pulse is shown as Fig. 5. However, the pulse at the present does not conform to any function model and just presents a nonlinear curve form. Therefore, in order to obtain stable peak waveform trajectory, we use cubic spline interpolation fitting for the pulse to obtain smooth results.
- According to the previous analysis, the nonlinear interpolation algorithm has a good processing result for the echo pulse. Considering the complexity of calculation and the difficulty of implementation, we choose cubic spline interpolation for processing.
- It is no doubt that the interpolation over fit the original data. This conclusion can be verified in FIG. 6, which represents the Lissajous figure of the interference result obtained after interpolation. The scattered points of the interference result after interpolation are more convergent, so that more consistent with the original data, and the quality of the interference signal is improved.
3、What does Figure 9 want to illustrate? What is the difference between the three pictures? Please explain in detail so that readers can better understand this study.
Response: Thank you very much for your valuable advice.
- The improvement effect of the proposed method on demodulation signal can be directly explained by time domain signals, but the results cannot be quantified. In order to qualitatively analyze the improvement effect of the proposed algorithm on the integrity of demodulation signal. In Fig.9, we compare the power spectral density (PSD) of demodulated signal at 10Hz obtained by three different methods.
- The PSD obtained by direct demodulation corresponds to Fig. 9(a). The method proposed in this paper is that the pretreatment of demodulation signal is mainly realized by two steps: first, the original reflected pulse array is processed by cubic spline interpolation (PSD of demodulation result obtained by this step corresponds to Fig. 9(b)). Then, the peak trajectory of the smoothed pulse array is obtained and linear compensation is performed (PSD of the demodulation result obtained in this step corresponds to Fig. 9(c)).
- We added the description to the text.
4、There are too many small and clerical mistakes in the manuscript, such as in abstract "…improved by approximately 10 dB@10 …", in page 2 "…phase change is less then \pi…" and in Figures 7 and 8 "Tims".
Response: We agree with the reviewer’s opinion, and thank you for the suggestions.
- Some contents of the abstract have been revised, “…and the signal to noise ratio is improved by approximately 10 dB at 10 Hz.”.
- In Page 2, “However, when the phase change is less than π due to the mildly intrusive…”.
- The abscissa of Fig. 7 and Fig. 8 has been modified.
- Besides, we revise the full text in detail, and all of revisions to the manuscript have been marked up using the “Track Changes” function.
